# Overview of Cellular Therapeutics Clinical Trials: Advances, Challenges, and Future Directions

**DOI:** 10.3390/ijms26125770

**Published:** 2025-06-16

**Authors:** Meizhai Guo, Bingyi Zheng, Xiaoling Zeng, Xueting Wang, Chi-Meng Tzeng

**Affiliations:** Translational Medicine Research Center, School of Pharmaceutical Sciences, Xiamen University, Xiamen 361102, China; guomeizhai@126.com (M.G.); leozheng2023@163.com (B.Z.); kclcat@163.com (X.Z.); xuetingw1991@163.com (X.W.)

**Keywords:** cellular therapeutics, MSC, CAR-T, degeneration diseases, cancer, traditional Chinese medicine

## Abstract

Cellular therapeutics, encompassing stem cell-based regeneration and engineered immune cell platforms, have demonstrated efficacy in treating degenerative diseases, immune-related diseases, and oncology. However, low engraftment rates and limited long-term efficacy remain critical translational barriers. This review compiled clinical projects on cell therapy in China over the past five years (over 1200 patients across 172 clinical trials) to highlight its rapid development in recent years and illustrate the directions of indications for application. This review also analyzes published clinical achievements all over the world, revealing significant therapeutic improvements in degenerative disorders (40–60% improvement in Western Ontario and McMaster Universities Osteoarthritis Index (WOMAC) scores and oncology (78% ctDNA clearance, *p* < 0.001)). We propose integrating traditional Chinese medicine (TCM) bioactive compounds to enhance cell viability via C-X-C motif chemokine receptor (CXCR4) upregulation and mitochondrial biogenesis. Despite mechanistic insights, translational barriers include limited TCM validation (72% lacking single-cell omics) and regulatory misalignment. Future efforts should prioritize randomized trials and standardized TCM-cell therapy protocols to bridge discovery and clinical translation.

## 1. Introduction

According to statistics, over 20 million new cancer cases occur globally each year, with about 50% of patients facing a risk of death due to drug resistance or cancer recurrence [1]. However, the efficacy of traditional chemotherapy and targeted drugs has nearly reached its limit, often accompanied by severe side effects. Cell therapy, which modifies or utilizes autologous/allogeneic cells to achieve precise targeting or tissue regeneration, has shown significant efficacy in several early clinical trials. Nonetheless, its long-term safety, industrial bottlenecks, and ethical controversies still need to be addressed through rigorous clinical trials.

Cellular therapeutics represents a paradigm shift in precision medicine, leveraging viable human-derived cells for tissue regeneration and immune modulation. Governed by stringent ethical frameworks (e.g., World Medical Association Declaration of Helsinki) and regulated as advanced therapy medicinal products (ATMPs), these therapies require pharmaceutical-grade manufacturing and risk-adapted clinical trial designs [2]. Two dominant modalities drive the field: stem cell-based strategies (multipotent differentiation, paracrine signaling, immunomodulation) and engineered immune cell platforms (e.g., chimeric antigen receptor T cells (CAR-T)). Applications span degenerative diseases, oncology, and regenerative medicine, supported by emerging evidence in epigenetic rejuvenation and immune reprogramming [3]. This review aims to summarize and organize clinical trials of mesenchymal stem cells (MSCs) and immune cell therapies in China, manifesting their safety and efficacy, as well as breakthroughs in treating complex diseases. It also explores the biological molecular mechanisms behind their effects and investigates potential new advancements in combined traditional Chinese medicine and cell therapies.

### 1.1. Challenges and Solutions in the Practice of Cell Therapy Clinical Trials

Cell therapy methods are still being continuously explored and improved. Clinical trials of cell therapy developed and registered in accordance with relevant pharmaceutical management regulations must comply with the requirements of the Good Clinical Practice for Drug Trials. This includes establishing a clinical trial quality management system by the sponsor and conducting risk-based quality management. Even when sponsors fully fulfill their responsibilities and develop well-designed protocols, there are still many challenges for clinical trial institutions and principal investigators in managing and conducting clinical trials.

#### 1.1.1. Selection of Study Populations

Cell therapy is often used for currently difficult-to-treat diseases with poor efficacy, such as blood system tumors. The severity of adverse reactions is high, with cytokine release syndrome (CRS) potentially causing high fever, coma, or even death. Additionally, the long-term safety of gene-modified cell therapy products is unclear [4], leading to significant uncertainty in risk-benefit assessments [5]. Cell therapy is still in its early stage of development, with considerable risks. In clinical trials, the risk–benefit ratio should be thoroughly considered. Patients should not directly opt for cell therapy as their first treatment option without undergoing standard treatment [6]. Patients whose condition has progressed after standard radiotherapy, chemotherapy, or targeted therapy have poorer physical conditions, and researchers should carefully evaluate whether these participants can tolerate cell therapy. Patients usually lack comprehensive understanding of cell therapy and its potential side effects. Researchers must fully inform participants about the trial process and risks during the informed consent process and appropriately address any questions raised by the participants [5].

Cell therapy should also consider minors, conducting trials only after dose exploration and initial safety and efficacy validation have been completed in adult subjects, with separate clinical trial protocols and plans for adverse event prevention and management.

#### 1.1.2. Cell Source and Collection

The cell collection techniques for clinical trials of cellular therapies have become relatively mature and have a minor impact on the overall quality of clinical trials. However, cell collection is the first step in ensuring the quality of the cells to be reinfused. Researchers should thoroughly evaluate the impact of cell collection on the patient’s health status [5]. Appropriate software and hardware collection conditions can ensure the smooth progression of subsequent processes. Therefore, the study protocol must clearly define the bridging chemotherapy and pre-treatment protocols before treatment, specifying the cell collection conditions, including the patient’s health status, specific numerical requirements for various test and examination indicators, and the number of allowable retests and time range, to maximize the patient’s treatment possibilities within a controllable risk range. The research team needs to complete cell collection in an environment that meets the criteria and transport it via cold chain logistics to the designated location according to the protocol requirements to proceed with subsequent operations [7].

For allogeneic source cell preparations, such as MSCs and umbilical cord blood stem cells, collection requires informed consent from the donor, and storage conditions and locations must comply with relevant regulations. The principal investigator should confirm the cell source, preparation process, and compliance of the cell preparation before conducting clinical trials [8]. During ethical review, apart from reviewing clinical trial-related materials, it should also review documents such as the donor’s informed consent form, storage institution’s storage qualification documents, and safety support documents for cell preparation and expansion. Allogeneic source cells are often stem cell preparations, and their storage time may be long, posing a risk of microbial contamination during storage and preparation. For gene-modified cell preparations, the method of gene editing affects safety risks. When discussing the clinical trial protocol with the sponsor, the principal investigator should fully consider safety risks and corresponding clinical tests and examinations, and improve follow-up test arrangements, set reasonable follow-up periods, and if necessary, arrange long-term follow-ups to confirm the long-term and potential toxicity of gene modification and gene introduction [9].

#### 1.1.3. Transport and Storage of Cell Preparations

Cell preparations require higher standards for storage and transportation compared to regular pharmaceuticals or medical devices, and should comply with domestic and international guidelines for the transport of cell preparation products [10]. Clinical trial institutions should communicate with the sponsor about the storage requirements of the cell product before receiving it [11]. The pharmacy staff should learn and familiarize themselves with the corresponding conditions in advance and reserve space and equipment that meet the requirements to ensure proper storage of the cell product after receipt. Upon receiving the cell product, in addition to verifying the quantity, specifications, and delivery temperature, it is also necessary to confirm that the temperature during transportation meets the storage requirements of the product before accepting it. The sponsor should investigate cell stability in advance, clearly define the extreme conditions under which the cells can be used, and if there is a brief temperature excursion during storage or transportation, promptly determine whether the cells can still be used to avoid delaying patient treatment and trial progress. Researchers should make the necessary preparations (such as warming the cells) before administering them, and keep detailed records.

#### 1.1.4. Monitoring and Management of Adverse Events in Cell Therapy Clinical Trials

Adverse event management is crucial in cellular therapy clinical trials. Cell therapy has shown good efficacy, but its treatment mechanism is complex and its biological effects are not well understood [8]. Chemical drugs and biologics have clear mechanisms of action, allowing safety risks identified in preclinical data to predict clinical trial risks and prepare accordingly. Different types of cell therapies have varying safety risks; for example, with chimeric antigen receptor T-cell therapy, common severe adverse events are immune-related, including CRS and immune effector cell-associated neurotoxicity syndrome (ICANS). The type and severity of adverse events occurring at different stages after cell infusion vary, necessitating graded and categorized management [12].

Researchers cannot predict clinical trial safety risks based on preclinical safety data of human-derived cell products. Therefore, the principal investigators and research teams need to have substantial knowledge and experience in cell therapy. They must undergo thorough training before starting the study to properly handle adverse events related to cell therapy, especially severe adverse events, and identify and intervene as early as possible [7]. Additionally, participants in cell therapy clinical trials are often patients with malignant diseases, who have poor health conditions, leading to a high frequency of adverse events during the trial, complex concomitant medication, and difficulty in assessing the correlation between adverse events and the clinical trial. Sponsors should conduct pharmacovigilance work, systematically analyze the correlation between adverse events and the clinical trial and promptly inform all principal investigators to prepare contingency plans. Researchers should regularly participate in training related to cell therapy clinical trials. During the training, they should not only learn about knowledge relevant to clinical trial practice but also exchange experiences in conducting clinical trials, preventive measures for adverse events, and handling procedures among researchers in the field, thereby improving the execution and quality of clinical trials.

## 2. Cell Therapy Landscape in China

The clinical advancement of cellular therapeutics in China has been driven by two landmark discoveries: the differentiation potential of human embryonic stem cells (hESCs) reported in 1998 [13] and the reprogramming of induced pluripotent stem cells (hiPSCs) in 2007 [14]. Cell therapy clinical research initiated by researchers often suffers from insufficient scientific design, inadequate ethical review, low reliability of research data, and weak homogeneity in research quality across different centers. This article focuses on discussing the relevant content of cell therapy clinical trials developed and registered according to relevant pharmaceutical management regulations. To comprehensively understand the development and registration of clinical trials for cell therapies according to relevant pharmaceutical regulations, if searching with the keyword ‘cells’, the scope is broad, covering cell therapy clinical trials, vaccine clinical trials, and various chemical drugs, protein biologics, etc. By observing the characteristics of the target trial titles, the search terms ‘stem cells’ and ‘cell injection solution’ were chosen. As of 14 May 2025, 168 and 150 records were found, respectively, on the ‘Drug Clinical Trials Registration and Information Disclosure Platform’ (http://www.chinadrugtrials.org.cn/index.html (accessed on 14 May 2025)). After excluding 33 duplicate trials, 111 unrelated trials, and 2 ‘voluntary suspension’ cases, the remaining 172 records are all cell therapy clinical trials. There are 57 trials related to mesenchymal stem cell therapy, and another 115 trials related to immunotherapy with a notable underrepresentation of pediatric populations (2.91%, n = 5 trials). This is presumably due to heightened safety and ethical hurdles for pediatric cell therapy trials (accessed on 14 May 2025), coupled with parental reluctance due to limited public awareness of clinical research. Also, commercial disincentives for pharmaceutical companies, including the relatively small pediatric market size, complex trial designs, and higher costs, deter investment compared to adult-focused therapies. Recently, the National Medical Products Administration (NMPA) has prioritized pediatric drug reviews, establishing a green channel for accelerated approval of pediatric drugs and therapies, which now extends to cell therapies [15]. Immune cell therapies were predominantly represented by CAR-T therapies (55.81%, n = 96/172), reflecting their dominance in hematologic malignancy research (Figure 1).
Stem cell trials (*n* = 57): Degenerative disorders (Figure 2) accounted for 59.65% (*n* = 34), including diabetes complications, osteoarthritis, and ischemic stroke etc.

**Figure 2 ijms-26-05770-f002:**
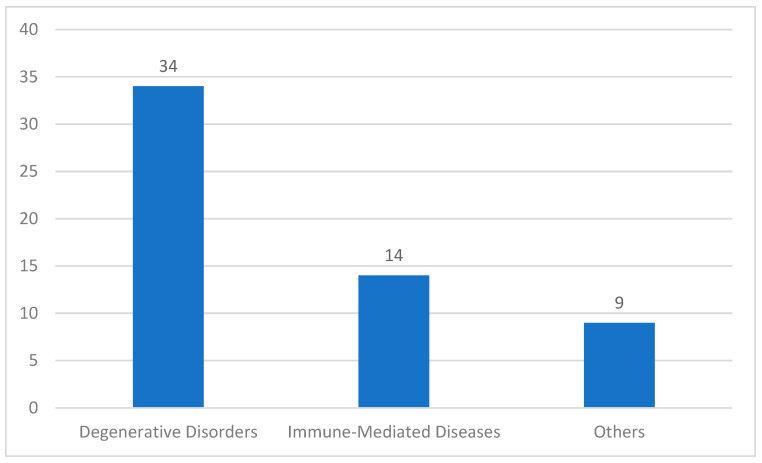
Recent clinical trials of stem cell therapies in China (classified by indication). There are 34 clinical trials for degenerative disorders, 14 related to immune-mediated diseases, and 9 for other complex illnesses.

Immune cell trials (n = 115): Hematologic malignancies were targeted in 40.87% (n = 47), with CAR-T therapies focusing on CD19+ B-cell malignancies (63.83%, n = 30/47).In the analysis of stem cell types (Figure 3), human stem cells have diverse sources. Umbilical cord mesenchymal stem cells (UCMSC) account for 35 items (44.2%), as they are easily accessible and have broad applications [14].

**Figure 3 ijms-26-05770-f003:**
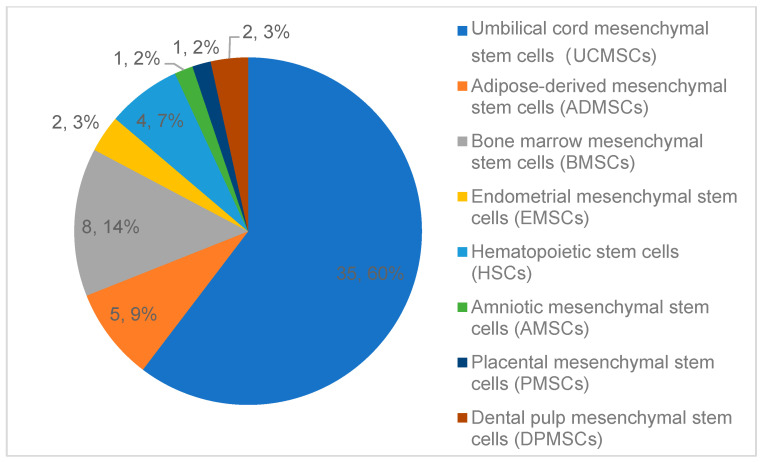
Recent clinical trials of stem cell therapies in China (classified by cell type). The most common sources are UCMSCs (35 clinical trials), BMSCs (eight clinical trials), ADMSCs (five clinical trials), HSCs (four clinical trials), EMSCs and DPMSCs (two clinical trials each), AMSCs (one clinical trial), and PMSCs (one clinical trial).

Phase I/II trials constituted 88.95% (n = 153/172) of the analyzed studies, highlighting an emphasis on preliminary safety assessments. Cumulative cell doses administered exceeded 190 billion across more than 1200 patients, with a median follow-up duration of 18 months (interquartile range [IQR] 12–24) [3]. The cell therapy method in which the treatment was associated with most adverse events was Olfactory ensheathing cell and BMSC combination therapy (55%), and the lowest level of adverse events was with embryonic stem cell therapies (2.33% of patients) [16]. In a meta-analysis, the total prevalence of adverse events in cell therapy was 19% and the highest pulled effect size belonged to urinary tract and localized adverse events; for example, the most common adverse events were transient backache and meningism (90%) and cord malacia (80%) in MSCs-based clinical trials of spinal cord injury [16]. The most frequently reported adverse event is transient fever, predominantly associated with intravenous infusion. This fever typically occurs within 24 h post-infusion and resolves spontaneously within 48 h without requiring intervention [17]. Administration-site reactions, including localized pain, swelling, or erythema, occur in up to 12% of patients receiving intra-articular or intralesional injections. Severe adverse events (SAEs), such as thromboembolism, infections, and organ-specific toxicities, are infrequent. These SAEs are often attributable to procedural factors or patient comorbidities rather than inherent properties of the cells themselves [18,19]. Also, the total prevalence of adverse events in 14 cell therapy methods was 18% and four cell types (neural stem cell, bone marrow hematopoietic stem cell, embryonic stem cell, and UCMSC) had the most effect. None of the adverse events were reported at (death) grading scales 4 (life-threatening consequences) and 5 [16]. Notably, CAR-T therapies were associated with higher incidences of cytokine release syndrome (CRS: 68.2%) and immune effector cell-associated neurotoxicity syndrome (ICANS: 21.4%) [20].

## 3. Disease-Specific Clinical Advancements and Therapeutic Efficacy

### 3.1. Degenerative Diseases

#### 3.1.1. Osteoarthritis (OA)

Osteoarthritis, characterized by progressive cartilage degradation, synovial inflammation, and subchondral bone remodeling, affects over 500 million individuals globally, with aging and obesity as key risk factors. Molecular pathogenesis involves dysregulated chondrocyte metabolism, mitochondrial dysfunction, and elevated pro-inflammatory cytokines (e.g., IL-1β, TNF-α), which drive extracellular matrix breakdown. MSCs mitigate oxidative stress, cellular senescence, and apoptosis by restoring dysfunctional mitochondria in chondrocytes, thereby facilitating cartilage regeneration. Exosomes play a pivotal role in mediating the therapeutic effects of MSCs (Figure 4) [21]. MSCs mitigate OA progression via paracrine secretion of anti-inflammatory cytokines (e.g., IL-10, TGF-β), exosomal miRNA-mediated inhibition of catabolic pathways (e.g., MAPK/NF-κB), and mitochondrial transfer to restore chondrocyte bioenergetics. A study suggested that hUC-MSC-derived exosomal miR-199a-3p alleviates OA by inhibiting the mitogen-activated protein kinase 4/nuclear factor-kappaB signaling pathway. The present findings suggest that miR-199a-3p delivery by hUC-MSC-Exos may be a novel strategy for the treatment of OA [22].

Numerous clinical trials have demonstrated that stem cell therapy significantly alleviates symptoms with minimal adverse effects (Table 1). Clinical trials demonstrate intra-articular MSC administration (10–100 × 10^6^ cells) improves WOMAC scores by 40–60% at 12 months [23], with high-dose cohorts showing enhanced cartilage regeneration on MRI [24]. Furthermore, it offers long-term therapeutic benefits by improving joint function. However, challenges remain: (1) therapeutic outcomes vary among individuals, with some patients showing no response; (2) the high cost of treatment often necessitates multiple administrations for efficacy; and (3) as shown in Table 1, ADMSCs that exhibit superior chondrogenic differentiation capacity due to their high expression of transcription factors SOX9 and COL2A1 are predominantly selected [25]. Future directions include combinatorial strategies with hydrogels or platelet-rich plasma (PRP) to enhance MSC retention and standardized protocols for exosomal miRNA profiling to predict patient-specific outcomes. Chinese researchers, specifically from Peking University Third Hospital, developed an innovative hydrogel encapsulation technology (Mg^2^⁺/Dimethyloxalylglycine-preconditioned hyaluronic acid methacrylate-phenylboronic acid (HAMA-PBA)) to enhance MSC therapy for osteoarthritis [26]. This self-healing, tissue-adhesive hydrogel significantly improved MSC survival by three-fold within the hostile joint environment and sustained therapeutic efficacy for over 12 months, evidenced by a 50% increase in cartilage repair markers (COL-II, GAG) secretion [26].

#### 3.1.2. Neurodegenerative Disorders

Neurodegenerative diseases, including Parkinson’s disease (PD), Alzheimer’s disease (AD), and ischemic stroke (IS), collectively account for 12% of global disability-adjusted life years [45]. PD pathogenesis involves α-synuclein aggregation and dopaminergic neuron loss, while AD is driven by Aβ plaques, tau hyperphosphorylation, and neuroinflammation [46]. In stroke, ischemia-reperfusion injury triggers excitotoxicity, oxidative stress, and blood–brain barrier disruption [47]. A study discusses the molecular mechanisms via which flavonoids and MSC therapy influence synaptic plasticity as well as their therapeutic potential in neurodegenerative diseases. Flavonoids modulate key signaling pathways such as MAPK/ERK and PI3K/Akt/mTOR to support neuroprotection, synaptic plasticity, and neuronal health, while also influencing neurotrophic factors (BDNF, NGF) which are secreted by MSCs and their receptors (TrkB, TrkA) [48]. By means of three molecular pathways (prostaglandin E2 (PGE2)), tumor-necrosis-factor-inducible gene 6 protein (TSG-6), and progesterone receptor (PR) and glucocorticoid receptors (GR), MSCs induce the activation of macrophages/microglia and drive them to polarize into the M2 phenotypes, which inhibits the release of pro-inflammatory cytokines and promotes tissue repair and nerve regeneration [49]. The therapeutic mechanisms of MSCs in IS include the regulatory effects of MSCs on microglia, the dual role of MSCs in astrocytes that produce excitatory neurotransmitters to protect neurons and produce TNF-α to balance inflammation, how MSCs connect innate and adaptive immunity, the secretion of cytokines by MSCs to counteract apoptosis and MSC apoptosis, the promotion of angiogenesis by MSCs to favor the restoration of the blood–brain barrier, and the potential function of local neural replacement by MSCs [50]. The findings indicate that Wilms’ tumor 1-associated protein (WTAP) depletion can enhance the alleviative effects of MSCs Exo on oxygen–glucose deprivation/reoxygenation-triggered cellular damage in SK-N-SH cells by downregulating receptor protein RPL9 [51].

Trans-European PD trials reported sustained dopaminergic neuron engraftment (42% fluorine-18 fluorodopa positron emission tomography (18F-DOPA PET) signal increase at 48 months) but limited motor improvement beyond 24 months [52]. The results of IS clinical trials are promising, in the sense that most methods used for stem cell transplantation appear to be safe. Table 2 shows that intravenous or intra-arterial transplantation is preferred in the acute phase, where the aim is to ameliorate systemic and local inflammation and cell engraftment is not required. Alternatively, intracerebral transplantation is preferred in the chronic phase, where cell engraftment is considered the objective of cell therapy [53]. However, optimal parameters including the choice of cell type, cell dose, and patient characteristics remain elusive and further research is needed to maximize the effects of the proposed methods [53]. Future research must prioritize biomaterial scaffolds to enhance neuronal integration and CRISPR-edited MSCs overexpressing neurotrophic factors to amplify therapeutic potency.

#### 3.1.3. Type 2 Diabetes Mellitus (T2DM) and Complications

Type 2 diabetes mellitus (T2DM), which accounts for 90–95% of all diabetes cases [96], is a chronic metabolic disorder characterized by insulin resistance and impaired insulin secretion. Globally, approximately 415 million adults are affected by diabetes, with an additional 318 million at high risk due to impaired glucose tolerance [97]. T2DM is a major risk factor for ischemic heart disease, stroke, chronic kidney disease, and adult-onset blindness [98]. Its pathogenesis involves complex interactions between genetic and environmental factors, leading to chronic low-grade inflammation, pancreatic beta cell dysfunction, and insulin resistance [99]. MSCs possess differentiation potential, immunosuppressive properties, and anti-inflammatory effects, making them a promising therapeutic candidate for T2DM [94]. MSCs can differentiate into insulin-producing cells (IPCs), promote the regeneration of pancreatic islet beta cells, protect endogenous islet cells, and improve insulin resistance, thereby exerting a positive impact on T2DM [100].

Numerous clinical studies have demonstrated the efficacy of MSCs in treating T2DM. For instance, in 2019, Esteban et al. found that combining BMSCs with hyperbaric oxygen therapy effectively reduced HbA1c levels in T2DM patients for up to one year [94]. However, the duration of MSC treatment effects remains a concern [101]. Some studies have shown that HbA1c levels decrease transiently but do not continue to decline over the long term [102]. Representative cases include a patient treated with BMSCs who achieved significant improvements in glycemic control and insulin resistance [103]. Future T2DM therapies may focus on MSC-based approaches, such as optimizing the source and dose of MSCs, developing personalized treatment regimens (changing the method of infusion, administering MSCs directly into the lymph nodes around the pancreas [104]), combining with gene therapy (silencing the MST1 gene in embryonic stem cells (ESCs) enhanced differentiation into IPCs that increased insulin secretion and improved glucose responsiveness in T2DM rat models, with normoglycemia maintained for 6+ weeks post-transplant [105]), and enhancing the understanding of the molecular mechanisms underlying MSC treatment to improve its efficacy and safety. Additionally, strategies to prevent the recurrence of diabetes post-treatment should be explored.

In diabetic nephropathy applications, MSCs significantly improved mitochondrial function in renal tubular epithelial cells by upregulating peroxisome proliferator-activated receptor gamma coactivator 1α (PGC-1α) expression, regulating mitochondrial fusion and fission proteins, reducing mitochondrial reactive oxygen species (ROS) production, and suppressing NACHT, LRR, and PYD Domains-Containing Protein 3 (NLRP3) inflammasome activation [106]. Furthermore, MSC treatment reduced the levels of pyroptotic markers, such as IL-18, and exhibited a marked anti-fibrotic effect in the long term. These findings suggest that MSCs not only repair kidney injury but also offer long-term protection against fibrosis [106,107].

#### 3.1.4. Cardiac Tissue Engineering Milestones

Recently, tunneling nanotubes (TNTs)—a novel type of long-distance intercellular connective structure—have been identified between MSCs and cardiomyocytes (CMs). The results demonstrate that isoproterenol (ISO) promotes the formation of TNTs, particularly between MSCs and CMs, and induces changes in the morphology of TNTs (thickening and lengthening). Additionally, MSCs transmitted Cx43 to CMs via TNTs, which contributes to the alleviation of ISO-induced myocardial hypertrophy. These results suggest that TNTs represent an important mechanism in MSC-mediated therapy for myocardial hypertrophy [108].

Cardiac cell therapy has progressed to Phase III validation. The trial (NCT01768702) demonstrated significant reductions in left ventricular end-systolic volume (−15.4 mL, *p* = 0.02) and increased 6-min walk distance (+34.9 m, *p* = 0.03) in ischemic cardiomyopathy patients receiving cardiopoiesis-guided mesenchymal stromal cells [109]. Innovations in electromechanical coupling, such as connexin-43 overexpression, improved engraftment rates by 2.3-fold in porcine myocardial infarction models, mitigating historical arrhythmogenicity risks [110].

#### 3.1.5. Hepatic Regeneration Paradigms

LyGenesis’ ectopic hepatogenesis platform restored albumin levels to normal in 42% of end-stage liver disease patients following lymph node delivery of allogeneic hepatocytes (50 million cells; 6-month follow-up) [111]. Phase IIa trials (150 million cells) now employ 99mTc-mebrofenin scintigraphy to quantify metabolic activity in ectopic liver tissue [111]. Preclinical porcine models revealed a 1:30 donor-to-recipient scaling ratio via Notch/epidermal growth factor (EGF) co-stimulation, offering a scalable solution to donor shortages. MSCs can regulate signaling pathways, including hepatocyte growth factor/c-Met, Wnt/beta (β)-catenin, Notch, transforming growth factor-β1/Smad, interleukin-6/Janus kinase/signal transducer and activator of transcription 3, and phosphatidylinositol 3-kinase/PDK/Akt, thereby influencing the progression of liver fibrosis and regeneration [112].

### 3.2. Immune-Mediated Diseases

#### 3.2.1. Autoimmune Disorders

Rheumatoid arthritis (RA) and systemic lupus erythematosus (SLE) are characterized by dysregulated adaptive immunity, including Th17/Tfh hyperactivation and autoantibody production. In RA, synovial fibroblast proliferation and IL-17/IL-21-driven inflammation perpetuate joint destruction. MSCs suppress pathogenic T cells via IDO/kynurenine-mediated apoptosis, promote Treg differentiation through TGF-β/IL-10, and inhibit B-cell maturation by downregulating BAFF/APRIL. MSCs show potential in cases of autoimmune disease and organ transplantation due to their immune regulation and anti-inflammatory properties. Comprehensive sample analysis revealed dysregulation of FGL1/LAG-3 and PD-L1/PD-1 immune checkpoints in allogeneic heart transplantation mice and clinical kidney transplant patients. MSCs not only enrich FGL1/PD-L1 expression but also maintain the immunomodulatory properties of unmodified MSC extracellular vesicles. It is confirmed that FGL1 and PD-L1 on extracellular vesicles (EVs) are specifically bound to their receptors, LAG-3 and PD-1 on target cells [113].

A Phase I/II trial (NCT01547091) demonstrated intravenous umbilical cord MSCs (1 × 10^6^/kg) reduced disease activity scores in 28 joints (DAS28) in RA patients 1 year and 3 years after treatment compared to before treatment (*p* < 0.05), correlating with decreased anti-circular citrullinated peptide (anti-CCP) titers [114]. In SLE, MSC infusion normalized CD4+/CD8+ ratios and reduced proteinuria by 60% in a 12-month follow-up (ChiCTR1800018084) [115]. Future strategies include engineering MSCs to overexpress anti-IL-6R nanobodies and utilizing single-cell omics to identify patient-specific immune signatures for precision therapy [116].

Vertex Pharmaceuticals’ VX-880 allogeneic islet cell therapy achieved insulin independence in 64% of T1DM recipients (*p* < 0.001 vs. controls), likely due to restored pancreatic β-cell mass [104]. Complementary adoptive transfer of regulatory T cells (Tregs) preserved fasting C-peptide levels via CTLA-4/IgG Fc-mediated suppression of autoimmune destruction, demonstrating the synergy between cellular replacement and immunomodulation [117].

#### 3.2.2. Inflammatory Conditions (COVID-19, Chronic Obstructive Pulmonary Disease (COPD))

Severe COVID-19 and COPD involve cytokine storms (IL-6, TNF-α) and alveolar macrophage dysfunction, leading to acute respiratory distress syndrome (ARDS) and pulmonary fibrosis [118]. MSCs attenuate hyperinflammation via ACE2-mediated SARS-CoV-2 neutralization [119], mitochondrial donation to epithelial cells, and macrophage reprogramming via Galectin-1/TSG-6 pathways [120].

In COVID-19 ARDS patients, intravenous MSC therapy (200 × 10^6^ cells) reduced mortality by 58% (REMAP-CAP trial) [121], while in COPD, MSC administration improved forced expiratory volume in one second (FEV1) by 12% at 6 months (NCT00683722) [122]. Challenges include transient engraftment and donor variability. Next-generation approaches focus on aerosolized MSC-EVs for targeted lung delivery [123] and CRISPR-Cas9-edited MSCs with enhanced anti-fibrotic (shRNA-TGF-β) and antiviral (IFN-λ1) properties [124].

### 3.3. Car-T Innovations

Chimeric antigen receptor (CAR) T-cell therapy has emerged as a potentially curative approach for hematological malignancies. The pooled analysis demonstrated an overall response rate of 75%, with a complete response achieved in 66% of patients. Moreover, 49% of patients demonstrated progression-free survival (PFS) with a median follow-up of 30 months, and 53% of patients achieved negative measurable residual disease (NMRD) remission [125]. Notably, few patients experienced CRS of grades 1–2; however, neurotoxicity was not described as a prevalent side effect [125]. DNA transposon-generated CD19 CAR T-cell therapy demonstrates promising efficacy in B-cell malignancies, with favorable safety profiles [126]. However, the outcomes of this meta-analysis underscore the need for further clinical development [127].

Off-the-shelf human pluripotent stem cell (hPSC)-derived CAR-NK therapies (e.g., Fate Therapeutics’ FT500) achieved 78% circulating tumor DNA (ctDNA) clearance in solid tumor cohorts (NCT04106167) through TIM-3/LAG-3 dual checkpoint knockout [128]. CRISPR-engineered neoantigen-specific T cell receptor (TCR)-T constructs demonstrated 91% tumor infiltration efficiency (PET-CT SUVmax ≥ 3.5) in Phase I trials, with no grade ≥ 3 cytokine release events [129]. Additionally, bioreactor-expanded megakaryocytes generated 3.5 × 10^11^ platelets per unit, achieving hemostatic equivalence to donor-derived products in thrombocytopenic models (Δ bleeding time −2.1 min, *p* = 0.01) [130]. Each product expressed a patient-specific neoTCR and was administered in a cell-dose-escalation, first-in-human phase I clinical trial (NCT03970382) (Figure 5). One patient had grade 1 cytokine release syndrome and one patient had grade 3 encephalitis. All participants had the expected side effects from the lymphodepleting chemotherapy. Five patients had stable disease and the other eleven had disease progression as the best response on the therapy [131]. neoTCR transgenic T cells were detected in tumor biopsy samples after infusion at frequencies higher than the native TCRs before infusion. This study demonstrates the feasibility of isolating and cloning multiple TCRs that recognize mutational neoantigens. Moreover, simultaneous knockout of the endogenous TCR and knock-in of neoTCRs using single-step, non-viral precision genome-editing are achieved. The manufacture of neoTCR engineered T cells at clinical grade, the safety of infusing up to three gene-edited neoTCR T cell products and the ability of the transgenic T cells to traffic to the tumors of patients are also demonstrated [131].

## 4. Discussion and Future Perspectives

While early successes in cellular therapeutics centered on CNS and ocular applications, recent advances span immune, cardiac, and metabolic disorders. CAR-T and stem cell therapies now comprise 31% of clinical interventions, with marked efficacy in diabetes, epilepsy, and age-related macular degeneration. However, transitioning from Phase I/II to Phase III trials requires resolving many barriers including low engraftment rates, limited long-term efficacy, adverse reactions, unclear long-term safety, risk–benefit assessment difficulties, patient understanding, cell handling challenges, and ethical dilemmas. The most critical translational bottleneck confronting cell therapy in China is manufacturing standardization, primarily stemming from the inherent heterogeneity of cellular products (e.g., MSCs, iPSCs). Variations across donors, tissue sources, and production batches severely compromise quality control and therapeutic consistency. This variability results in unreliable clinical outcomes, limited scalability, and barriers to regulatory-compliant translation [132].

The integration of TCM-derived natural compounds with stem cell therapy represents an emerging strategy to overcome key limitations in regenerative medicine, such as low cell survival, uncontrolled differentiation, and insufficient endogenous stem cell pools [133]. The theoretical synergy arises from TCM’s holistic approach to systemic modulation, which aligns with stem cells’ capacity for multi-tissue regeneration [134]. TCM compounds epigenetically regulate stem cell niches in vivo by modulating signaling pathways (e.g., Wnt/β-catenin, MAPK/ERK), thereby enhancing the microenvironment for transplanted cells [135]. For example, safflower-derived palmitic acid upregulates CXCR4 expression in mesenchymal stromal cells (MSCs) [136], improving migration, while ginsenoside compound K (CK) enhances glucose transporter 1 (GLUT1)-mediated ATP production [137]. This mechanistic convergence provides a molecular foundation for combination strategies aimed at improving therapeutic precision.

Specific TCM-derived compounds significantly augment MSC functions, including proliferation, differentiation, and stress resistance, through defined molecular pathways. Icariin (from *Epimedium brevicornu*) stimulates osteogenic differentiation and suppresses adipogenesis in MSCs via miR-23a-mediated Wnt/β-catenin activation [138]. Curcumin (from *Curcuma longa*) synergizes with MSCs to mitigate neurological damage in ischemic stroke by activating the AKT/GSK-3β/β-TrCP/Nrf2 axis [139]. Resveratrol (found in grapes and peanuts) enhances MSC osteogenesis through Hippo/YAP and SIRT7/NF-κB pathways while inhibiting senescence [140]. These compounds exemplify TCM’s role as a “biocompatible adjuvant” that optimizes MSC performance for bone, neural, and vascular repair. Despite these mechanistic insights, translational barriers persist: (1) 72% of TCM effects lack single-cell multiomics validation (e.g., scRNA-seq/CITE-seq) [141]; (2) only 9% of TCM components meet Food and Drug Administration (FDA) botanical guidelines for quality-by-design (QbD), compared to 34% under NMPA standards [141]; and (3) no Phase III randomized controlled trials (RCTs) have evaluated TCM–cell therapy combinations (ClinicalTrials.gov: 0/126).

To address these gaps, we propose establishing standardized TCM-compound libraries with IC_50_ profiling and adopting organ-on-chip platforms for real-time interaction analysis. Concurrently, regulatory alignment between NMPA and FDA guidelines is critical to accelerate global validation. Future research must prioritize RCTs comparing combination therapies to monotherapies, ensuring rigorous safety and efficacy assessments. By bridging mechanistic discovery with clinical translation, the integration of TCM and cellular therapeutics may redefine regenerative and immunomodulatory medicine.

## Figures and Tables

**Figure 1 ijms-26-05770-f001:**
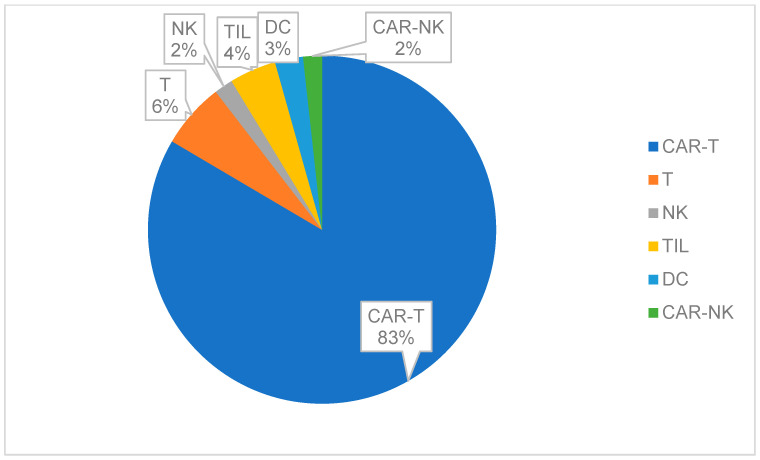
Recent clinical trials of immune cell therapies in China (classified by cell type): CAR-T account for 96 clinical trials, T cells seven clinical trials, natural killer cells (NK) two clinical trials, tumor infiltrating lymphocytes (TIL) five clinical trials, dendritic cells (DC) three clinical trials, and chimeric antigen receptor- natural killer cells (CAR-NK) two clinical trials.

**Figure 4 ijms-26-05770-f004:**
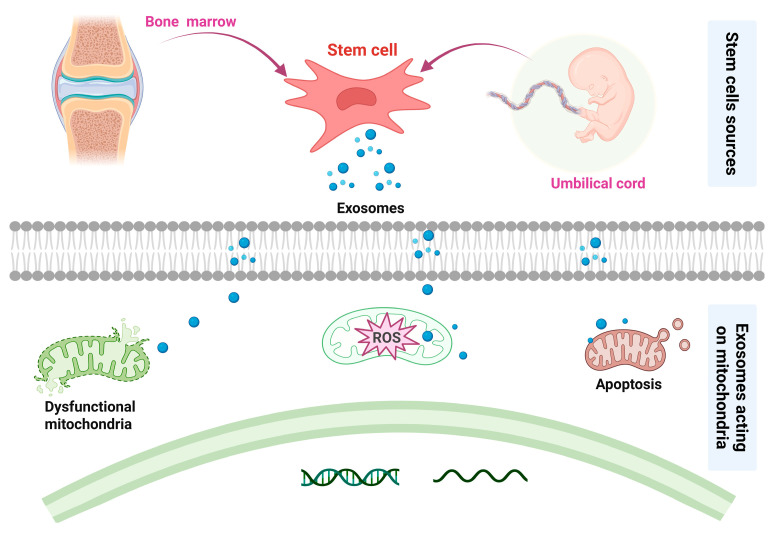
Potential mechanisms of MSC-based therapy for OA.

**Figure 5 ijms-26-05770-f005:**
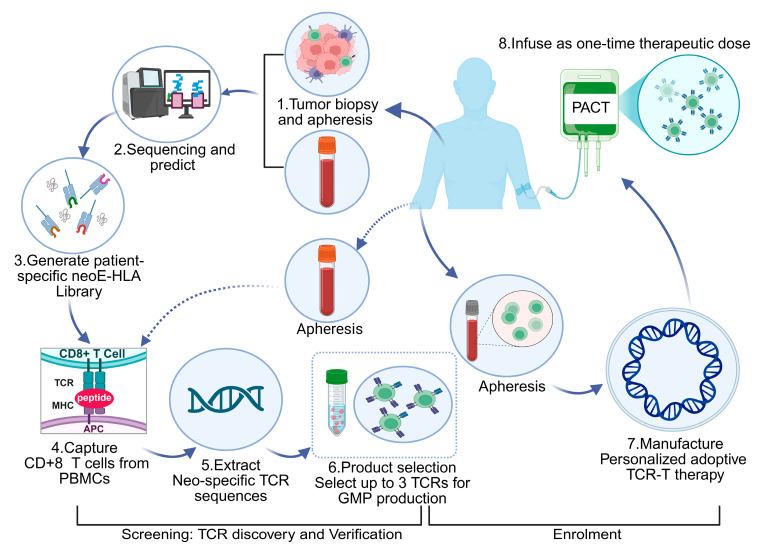
Generation of the neoTCR product for each patient.

**Table 1 ijms-26-05770-t001:** Clinical applications of MSCs in knee osteoarthritis.

Ref.	MSC Tissue Source	Cell Dose (×10^6^ Cells)	Number Treated	Time Points	Cartilage Outcomes
[27]	AD	10, 50, 100	12 (100 × 10^6^ dose); 3 (50 × 10^6^ dose); 3 (10 × 10^6^ dose)	0, 3, 6 months (MRl, X-Ray); 0, 6 months (arthroscopy histology)	Regeneration favoring high-dose group by MRl, arthroscopy, histology
[28]	BM	10, 100	10 (100 × 10^6^ dose); 10 (10 × 10^6^ dose); 10 (control)	0, 6, 12 months	Possible regeneration favoring high-dose group by X-Ray and MRI
[29]	AD	100	10 (single dose); 10 (two doses); 10 (control)	0, 12 months	Chondroprotection favoring two-dose group
[30]	BM	100	30 (MSC + PRP); 30 (control)	0, 12 months	No significant effects
[31]	AD	100	12 (MSC group); 12 (control)	0, 6 months	Chondroprotection; significant increase in defect size observed in saline- but not MSC-treated group by MRI
[32]	AD	10, 20, 50	6/dose group	0, 12, 24, 48, 72, 96 weeks	Regeneration favoring high-dose group with subsequent reduction in cartilage volume at 96 weeks
[33]	BM	1, 10, 50	4 (50 × 10^6^ dose); 4 (10 × 10^6^ dose); 4 (1 × 10^6^ dose)	0, 6, 12 months (MRI); 2, 6, 12, 24, and 48 weeks (ELISA)	Possible chondroprotection; no changes by MRl but catabolic biomarkers were significantly reduced and favored high-dose group
[34]	AD	50	12	0, 48 weeks	Regeneration indicated by reduced MOAKs articular cartilage pathology scores
[35]	AD	50	26/group	0, 24, 48 weeks	Regeneration in MSC group with degeneration in HA group
[36]	BM	40	12	0, 6, 12 months	Regeneration indicated by significant reduction in PCl scores at 6 and 12 months
[37]	BM	40	50	0, 12 months	Regeneration indicated by significant reduction in T2 values
[38]	BM	40	15	0, 6, 12 months	Regeneration indicated by significant reduction in T2 values at 6 and 12 months
[39]	BM	30.5	13	0, 6, 12 months	Regeneration indicated by increased cartilage thickness at 12 months
[40]	BM	25, 50, 75, 150	10 (for each MSC dose); 20 (control)	0, 6, 12 months (MRl); 0, 3, 6 months (X-Ray)	No significant effects
[41]	AD	100	3	0, 6 months (MRl); 0, 1 week, 1, 3, 6 months	Possible chondroprotection and regeneration by MRl and ELISA but small sample size
[42]	Placenta	50–60	10 (MSC); 10 (control)	0, 24 weeks	Regeneration indicated by increased cartilage thickness relative to baseline in MSC but not control group
[43,44]	AD	10, 20, 50	7 (10 × 10^6^ group); 8 (20 × 10^6^ group); 7 (50 × 10^6^ group)	0, 48 weeks	Regeneration favoring high-dose group

AD = Adipose Derived; BM = Bone Marrow Derived.

**Table 2 ijms-26-05770-t002:** Published clinical trials using stem cells for ischemic stroke.

Ref.	Country	Cell Source	Dose	Route	Transplant Timing	Treated Patient Number (Control)	Major Outcome
Acute							
[54]	USA	BM	4–6 × 10^8^	IV	1–3 D	10	Showed good neurological recovery
[55]	USA	BM	1.2 × 10^8^	IV	1–2 D	65 (58)	No difference
[56]	USA	UC	1.2 × 10^6^ (CD34+)	IV	3–9 D	10	Safe
[57]	Brazil	BM	5–6 × 10^7^	IA	3–10 D	20	30% of patients showed satisfactory clinical outcome
[58]	Spain	BM	1.6 × 10^8^	IA	5–9 D	10 (10)	No difference
[59]	Brazil	BM	3 × 10^7^	IA	9 D	1	Brain/liver/spleen uptake at 8 h
[60]	UK	BM	1–3 × 10^6^ (CD34+)	IA	1 W	5	Good recovery
[61]	China	UC & NPC	3 × 10^7^ (UC:IV), 1.5 × 10^7^ (UC:IT), 1.8 × 10^7^ (NPC:IT)	IV & IT	1 W	1	Showed some degree of neurological recovery
Sub-Acute							
[62]	India	BM	2–19 × 10^8^	IV	2–4 W	11	Favorable outcomes were mostly found in early treatment group
[63]	India	BM	5 × 10^7^	IV	3–4 M	1 (3)	Safe
[64]	Brazil	BM	2–5 × 10^8^	IV	1–3 M	5	Cells in brain were scarce (1%), IV (21%) showed high cell distribution in lung compared with IV (7%)
[65]	India	BM	2.8 × 10^7^	IV	18 D	59 (59)	No significant recovery compared with control
[66]	Japan	BM	2.5–3.4 × 10^8^	IV	7–10 D	12	Better NlHSS recovery compared with historical control
[67]	Korea	BM	1 × 10^8^	IV	1–2 M	5 (25)	Cell treatment group showed better neurological recovery than control
[68]	Korea	BM	1 × 10^8^	IV	2 M	16 (36)	Better recovery, less mortality within 5 years
[69]	Japan	BM	0.8–1.5 × 10^8^	IV	1–4 M	12	Recoveries were mainly seen 0–1 W from transplantation
[70]	China	BM	3 × 10^8^	IV	1 M	12 (6)	No neurological difference compared with control
[71]	France	BM	1 or 3 × 10^8^	IV	1–2 M	16 (15)	No overall change, but motor functional evaluations indicated improvement
[61]	China	UC	1.2 × 10^8^	IV	2 & 3 M	2	Showed some degree of neurological recovery
[72]	Brazil	BM	1–5 × 10^8^	IA	2–3 M	6	Cells were found in the brain after 2 h, but not after 24 h
[73]	Brazil	BM	1–5 × 10^8^	IA	2–3 M	6	Safe, but cells could not be seen 24 h after injection in 4 out of 6 patients
[64]	Brazil	BM	1–5 × 10^8^	IA	1–3 M	7	Cells in brain were scarce (1%), IA (41%) showed high cell distribution in liver compared with lV (13%)
[74]	Egypt	BM	1 × 10^6^	IA	2–4 W	21 (18)	lA treatment did not improve neurological recovery compared with control
[75]	India	BM	5 × 10^8^	IA	1–2 W	10 (10)	Good recovery was observed in treatment group (*p* = 0.06)
[76]	USA	BM	3 × 10^6^	IA	2–3 W	29 (17)	No statistical difference compared to control
[77]	China	UC & NPC	2 × 10^7^	IA	11–22 D	3	Showed neurological recovery in 2 out of 3 patients
[78]	Russia	Fetus neuronal cell	2 × 10^8^	IT	4 M	1	Cell treatment showed 33% increase in score
[61]	China	UC & NPC	3 × 10^7^ (UC:IV), 1.5 × 10^7^ (UC:IT), 1.8 × 10^7^ (NPC:IT)	IV&IT	2 W	1	Showed some degree of neurological recovery
Chronic							
[67]	India	BM	5 × 10^7^	IV	6–15 M	11 (9)	Significant improvement in mBl, but not in FM
[63,79]	India	BM	5–6 × 10^7^	IV	8–12 M	6 (6)	No significant difference compared with control up to 4 years
[80]	USA	BM	1 × 10^8^	IV	7 M–25 Y	36	Significant recovery was observed
[81]	India	BM	5–6 × 10^7^	IV	3 M–2 Y	20 (20)	mBl showed significant improvement
[82]	India	BM	6 × 10^7^	IT	4 M–12 Y	14	Showed recovery, but this study included hemorrhagic stroke
[83]	China	UC(CD34+)	1–3 × 10^7^	IT	1–7 Y	8	Patients showed recovery, but this may have been due to natural history
[84]	Russia	Fetus neuronal cell	2 × 10^8^	IT	8 M–1.5 Y	6 (6)	Cell treatment groups showed better recovery
[85]	USA	AD	N.D.	IT	1 Y	1	Stable
[86]	Cuba	BM	1–5 × 10^7^	IC	3–5 Y	3	Recovery compared with pre-operation was found
[87]	Taiwan	UC (CD34+)	3–8 × 10^6^	IC	6 M–5 Y	15 (15)	Statistically significant recovery
[88,89]	USA	BM	2.5, 5, 10 × 10^6^	IC	7–36 M	18	Neurological recovery (ESS, NIHSS, F-M test) was observed up to 2 years
[90]	USA	Fetus neuronal cell	2 × 10^6^ (n = 8) or 6 × 10^6^ (n = 4)	IC	7 M–5 Y	12	6 × 10^6^ showed better recovery than 2 × 10^6^
[91]	UK	Fetus neuronal cell	2, 5, 10, 20 × 10^6^	IC	1–4 Y	11	Neurological recovery (median NlHSS of 2) was observed
[92]	UK	Fetus neuronal cell	2 × 10^7^	IC	2 M–1 Y	23	Upper limb function recovered from baseline
[93]	USA	Fetus neuronal cell	5, 10 × 10^6^	IC	1–6 Y	18 (4)	No difference for neurological recovery (primary endpoint), but showed partial recovery in some tests
[94]	China	OEC	1 × 10^6^	IC	3 Y	1	Recovery in speech and gait
[94]	China	OEC & NPC	1 × 10^6^ & 2 × 10^6^	IC	5 Y	1	Recovery in motor function
[95]	USA	Fetus neuronal cell	2 × 10^7^	IC	1.5–10 Y	5	Slight recovery, but 2 patients exhibited adverse events (seizure and motor deficit)
[94]	China	OEC & NPC	1 × 10^6^ & 2 × 10^6^	IC & IT	1–20 Y	4	Recovery in gait
[61]	China	UC & NPC	3 × 10^7^ (UC:IV), 1.5 × 10^7^ (UC:IT), 1.8 × 10^7^ (NPC:IT)	IV & IT	10 M & 2 Y	2	Showed some degree of neurological recovery

BM = Bone Marrow Derived; UC = Umbilical Cord Derived; NPC = Neural Progenitor Cell; OEC = Olfactory Ensheathing Cell; IV = Intravenous; IT = Intrathecal; IC = Intracerebral; IA = Intra-Arterial.

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
