# Peer review of "Overview of Cellular Therapeutics Clinical Trials: Advances, Challenges, and Future Directions"

_ijms, 2025, doi:10.3390/ijms26125770_

Round 1
Reviewer 1 Report
Comments and Suggestions for Authors
This paper by Meizhai Guo et al. provides a review of the landscape of cellular therapeutics clinical trials, focusing particularly on advances, challenges, and future directions in China. The article positions cellular therapeutics as a paradigm shift in precision medicine, utilizing viable human-derived cells for tissue regeneration and immune modulation. The two main modalities discussed are stem cell-based strategies (including multipotent differentiation, paracrine signaling, and immunomodulation) and engineered immune cell platforms, such as chimeric antigen receptor T cells (CAR-T). These therapies are applied in various fields, including degenerative diseases, oncology, and regenerative medicine, supported by evidence in epigenetic rejuvenation and immune reprogramming. The authors summarize clinical trials involving mesenchymal stem cells and immune cell therapies in China to demonstrate their safety and efficacy, highlight breakthroughs, explore their biological mechanisms, and investigate potential advancements by combining them with traditional Chinese medicine (TCM). These combined elements make it very likely interesting for publication.
However, several concerns should be addressed to improve the manuscript:
-Could the authors provide more detail on the methodology used to identify and select the 172 cell therapy clinical trials from the Drug Clinical Trials Registration and Information Disclosure Platform? For example, were there specific inclusion/exclusion criteria applied beyond the search terms “stem cells” and “cell injection solution”?
-The authors report that 9.1% of cases experienced severe AEs (grade ≥3). Could they provide a breakdown of these severe adverse events by the type of cell therapy (MSC vs. Immune Cell) and, if possible, by specific indications?
-Beyond CRS and ICANS associated with CAR-T, could the authors elaborate on other types of severe adverse events observed in the analyzed trials, particularly those related to MSC therapies?
-The authors list several critical translational barriers, including low engraftment rates, limited long-term efficacy, adverse reactions, unclear long-term safety, risk-benefit assessment difficulties, patient understanding, cell handling challenges, and ethical dilemmas. In their opinion, which of these barriers are currently the most significant challenges facing cell therapy clinical translation in China, and why?
-The proposed integration of TCM bioactive compounds is a novel and interesting future direction. Could the authors discuss which specific TCM compounds, beyond palmitic acid and ginsenoside compound K, they believe hold the most promise for enhancing cell therapy outcomes, and why?
-The authors note the underrepresentation of pediatric populations in the analyzed trials (2.91%). What are the likely reasons for this, and are there specific initiatives or plans in China to increase clinical trial opportunities for pediatric patients requiring cell therapies?
-For diseases like Osteoarthritis or Type 2 Diabetes, where MSCs show promise but duration of effect is a concern, what strategies are being explored in China to enhance the persistence and long-term efficacy of the transplanted cells?
Author Response
Comment1: Could the authors provide more detail on the methodology used to identify and select the 172 cell therapy clinical trials from the Drug Clinical Trials Registration and Information Disclosure Platform?
Response1: Agree. We have, accordingly, modified the detail of methodology including key words”stem cells” and “cells injection solution”. Mention exactly where in the revised manuscript this change can be found—2. Cell therapy landscape in China, Paragraph 1, Page 4 and 164-168 lines.
Comment2: Could they provide a breakdown of these severe adverse events by the type of cell therapy (MSC vs. Immune Cell) ? Particularly related to MSC therapy.
Response2: Agree. We have, accordingly, discussed the adverse events of cell therapy, such as 1)transient fever, 2)administration-site reactions, 3)thromboembolism, 4)infections, and 5) organ-specific toxicities. Mention exactly where in the revised manuscript this change can be found—2. Cell therapy landscape in China, Page 6 and 218-233 lines.
Comment3: which of these barriers are currently the most significant challenges facing cell therapy clinical translation in China, and why?
Response3: Agree. We have, accordingly, discussed the most significant challenge in China that is manufacturing standardization of cell therapy products. Mention exactly where in the revised manuscript this change can be found—4. Discussion and Future Perspectives, Paragraph 1, Page 16 and 472-481 lines.
Comment4: Could the authors discuss which specific TCM compounds, beyond palmitic acid and ginsenoside compound K, they believe hold the most promise for enhancing cell therapy outcomes, and why?
Response4: Agree. We have, accordingly, analyzed the theoretical potential between TCM and cell therapy, and discussed the enhancing mechanism of TCM for cell therapy Mention exactly where in the revised manuscript this change can be found—4. Discussion and Future Perspectives, Page 17, 485-489 & 495-504 lines.
Comment5: The authors note the underrepresentation of pediatric populations in the analyzed trials (2.91%). What are the likely reasons for this, and are there specific initiatives or plans in China to increase clinical trial opportunities for pediatric patients requiring cell therapies?
Response5: Agree. We have, accordingly, discussed the reasons for limited clinical trials related to pediatric populations, such as safety , ethical hurdles, and Commercial disincentives. Mention exactly where in the revised manuscript this change can be found—2. Cell therapy landscape in China, Page 4, 180-187 lines.
Comment6: For diseases like Osteoarthritis or Type 2 Diabetes, where MSCs show promise but duration of effect is a concern, what strategies are being explored in China to enhance the persistence and long-term efficacy of the transplanted cells?
Response6: Agree. We have, accordingly, discussed strategies to enhance the efficacy of MSC therapy, for example 1) developing an innovative hydrogel encapsulation technology,2)developing personalized treatment regimens by gene editing technology. Mention exactly where in the revised manuscript this change can be found—3.1.1 Osteoarthritis (OA), Page 7, 269-276 lines and —3.1.3. Type 2 Diabetes mellitus (T2DM) and Complications , Page 13, 343-349 lines.
Reviewer 2 Report
Comments and Suggestions for Authors
On reviewing the review paper entitled “OverviewofCellularTherapeuticsClinicalTrials:Advances,Challenges,andFuture Directions”
Overall, the idea of the review is interesting and the review is generally comprehensive. However, there are many issues that should be addressed before acceptance for publication.
Abstract:
The abstract is well written an comprehensive. However, there are many abbreviations that are mentioned for the first time in the abstract without the full-length words.
However, I thought for the first while that the authors will take mainly about the studies in China. Meanwhile, the review reflect global studies. Therefore, the abstract must be rephrased to be aligned with the scope of the review.
Introduction
Line 55: It also explores the biological molecular 54 mechanisms behind their effects and investigates potential new advancements in combined traditional Chinese medicine and cell therapies: the font format need to be unified.
Section 1. from 1.1 to 1.4: is lacking enough supporting references and some parts are not supported by references. Even if the authors are displaying an discussing their opinions and comments, the basis of the discussion should be supported by valid references.
Figure1. The full-length of the abbreviations and clearer explanation of the figure should be written beneath it.
Figure 2: Clearer explanation of the figure should be written beneath it.
Figure 3: The same as figure 2
Kindly, write in the text the full-length word of WOMAC score, PRP, IS, TRANSEURO, 18F-DOPAPET, PGC-1α, NLRP3, EVs, anti-CCP, COPD, FEV1……..etc. This is a review that can help the reader to make a background about the stem cells therapeutic potential. Therefore, it should be clear to help young researchers. Therefore, Kindly revise the whole manuscript for the abbreviations.
Table 1: the abbreviations used are not mentioned in full-length word below the table.
LINE 255: the dual role of MSCs in astrocytes: What do the authors means by the dual role? Kindly, clarify for the reader.
The first section under the title of neurodegenerative diseases needs to be supported by more references. Large paragraphs are only supported by a single reference, while containing many details.
Each paragraph related to a certain point needs a reference. It seems that the authors take a certain part large paragraph from one paper as a literature and use it without even adding the original references used. For example, in the DM section, the authors talked about the prevalence and pathogenesis with just a single reference.
“PhaseIIItrial(NCT0318643)demonstratedintravenousumbilicalcordMSCs(1× 356 10^6/kg) reducedDAS28scoresby2.1points inrefractoryRApatients at 24weeks, 357 correlating with decreased anti-CCP titers. In SLE, MSC infusion normalized 358 CD4+/CD8+ ratios and reduced proteinuria by 60% in a 12-month follow-up 359 (ChiCTR1800018084). Future strategies include engineering MSCs to overexpress 360 anti-IL-6Rnanobodiesandutilizingsingle-cellomicstoidentifypatient-specificimmune 361 signaturesforprecisiontherapy.” ……This section needs a reference.
“SevereCOVID-19andCOPDinvolvecytokinestorms (IL-6,TNF-α)andalveolar 370 macrophagedysfunction, leading toARDS andpulmonary fibrosis.MSCs attenuate 371 hyperinflammation via ACE2-mediated SARS-CoV-2 neutralization, mitochondrial 372 donation to epithelial cells, andmacrophage reprogramming viaGalectin-1/TSG-6 373 pathways”……. This section needs a reference.
“DNAtransposon-generatedCD19CART-cell therapydemonstrates 389 promisingefficacyinB-cellmalignancies,withfavorablesafetyprofiles.However, the 390 outcomesofthismeta-analysisunderscoretheneedforfurtherclinicaldevelopment.” ”……. This section needs a reference.

Author Response
Comment1: The abstract is well written an comprehensive. However, there are many abbreviations that are mentioned for the first time in the abstract without the full-length words. Also, I thought for the first while that the authors will take mainly about the studies in China. Meanwhile, the review reflect global studies. Therefore, the abstract must be rephrased to be aligned with the scope of the review.
Response1: Agree. We have, accordingly, revised the abstract and added the abbreviations. Also, this review aims to highlight the Cell-Based Therapies in China and discuss the clinical progress around the world.Mention exactly where in the revised manuscript this change can be found—Abstract.
Comment2: It also explores the biological molecular 54 mechanisms behind their effects and investigates potential new advancements in combined traditional Chinese medicine and cell therapies: the font format need to be unified.
Response2: Agree. We have, accordingly, revised the font format . Mention exactly where in the revised manuscript this change can be found—1.Introduction.
Comment3: Section 1. from 1.1 to 1.4: is lacking enough supporting references and some parts are not supported by references.
Response3: Agree. We have, accordingly, added the reference to support this sections . Mention exactly where in the revised manuscript this change can be found—Section 1. from 1.1 to 1.4, Page 2-3.
Comment4: Figure1. The full-length of the abbreviations and clearer explanation of the figure should be written beneath it.; Figure 2: Clearer explanation of the figure should be written beneath it; Figure 3: The same as figure 2.
Response4: Agree. We have, accordingly, modified the figures for more details. Mention exactly where in the revised manuscript this change can be found—Figure1(Page 5), Figure2(Page 5),Figure3(Page 6).
Comment5: Therefore, Kindly revise the whole manuscript for the abbreviations, for example WOMAC score, PRP, IS, TRANSEURO, 18F-DOPAPET, PGC-1α, NLRP3, EVs, anti-CCP, COPD, FEV1.
Response5: Agree. We have, accordingly, modified the abbreviations for this manuscript.
Comment6: The dual role of MSCs in astrocytes: What do the authors means by the dual role? Kindly, clarify for the reader.
Response6: Agree. We have, accordingly, clarified the dual role of MSC. Mention exactly where in the revised manuscript this change can be found—3.1.2. Neurodegenerative Disorders, Page 10, 296-302 lines.
Comment7: The first section under the title of neurodegenerative diseases needs to be supported by more references. Also, in the DM section and other paragraphs.
Response7: Agree. We have, accordingly, added the reference for every sections. Also , we modified the format according to the IJMS.
Round 2
Reviewer 2 Report
Comments and Suggestions for Authors
I would like to thank the authors for responding to the raised points.